# ESVIO: Event-Based Stereo Visual-Inertial Odometry

**DOI:** 10.3390/s23041998

**Published:** 2023-02-10

**Authors:** Zhe Liu, Dianxi Shi, Ruihao Li, Shaowu Yang

**Affiliations:** 1College of Computer, National University of Defense Technology, Changsha 410005, China; 2Artificial Intelligence Research Center (AIRC), Defense Innovation Institute, Beijing 100166, China

**Keywords:** stereo visual-inertial odometry, event camera, sensor fusion, state estimation, event-inertial initialization

## Abstract

The emerging event cameras are bio-inspired sensors that can output pixel-level brightness changes at extremely high rates, and event-based visual-inertial odometry (VIO) is widely studied and used in autonomous robots. In this paper, we propose an event-based stereo VIO system, namely ESVIO. Firstly, we present a novel direct event-based VIO method, which fuses events’ depth, Time-Surface images, and pre-integrated inertial measurement to estimate the camera motion and inertial measurement unit (IMU) biases in a sliding window non-linear optimization framework, effectively improving the state estimation accuracy and robustness. Secondly, we design an event-inertia semi-joint initialization method, through two steps of event-only initialization and event-inertia initial optimization, to rapidly and accurately solve the initialization parameters of the VIO system, thereby further improving the state estimation accuracy. Based on these two methods, we implement the ESVIO system and evaluate the effectiveness and robustness of ESVIO on various public datasets. The experimental results show that ESVIO achieves good performance in both accuracy and robustness when compared with other state-of-the-art event-based VIO and stereo visual odometry (VO) systems, and, at the same time, with no compromise to real-time performance.

## 1. Introduction

Visual odometry (VO) is an emerging and hot research topic in the fields of robotics and computer vision [1], which aims to make the real-time estimation of camera poses and motion trajectory using only visual sensors (such as monocular camera [2], stereo camera [3], panoramic camera [4], depth camera [5], etc.). However, due to the limitation of imaging modalities, standard camera-based VO methods are not robust enough in some challenging scenarios, such as fast-moving, high dynamic range (HDR), rapid illumination changing, etc. In recent years, a new type of bio-inspired vision sensor, namely event camera, has gradually attracted the attention of researchers. It works differently from the standard camera and has independent pixels which can capture brightness changes individually and output this information as “event” asynchronously [6]. Compared to standard cameras, event cameras have the higher temporal resolution, wider dynamic range, lower latency, lower power consumption, and bandwidth [7,8]. With these advantages, researchers have developed a series of event-based VO methods [8,9,10,11,12] to provide correct camera state estimation in the challenging scenarios, as mentioned above. However, due to the hardware limitation of event cameras (such as noise and dynamic effects, low spatial resolution, etc.) and related research is still in the exploration stage, the event-based VO still faces the problems of low accuracy and low robustness in some scenarios.

In the traditional VO field, introducing a low-cost inertial measurement unit (IMU) to improve the accuracy and robustness of camera state estimation is a high-efficiency solution, which is called visual-inertial odometry (VIO) and has been a research highlight in the past decade [13,14,15,16,17,18]. For the event-based VO, integrating inertial measurement can also assist the system in estimating motion and recovery scale. At present, a few works have explored event-based VIO methods and achieved accurate state estimation results, such as [19,20,21,22,23,24]. However, two common challenges exist in the event-based VIO field: (1) how to process the novel event data and fuse them with IMU measurement to estimate camera motion. Most of the existing event-based VIO methods (except for [22]) convert event stream to event image frames and execute feature-based VIO pipelines based on these frames. These methods will extract and track specific visual features from consecutive event image frames. Then, they will minimize feature reprojection errors and IMU propagation errors to estimate camera motion. However, due to the working principle and hardware limitations of event cameras, most event image frames have the disadvantages of high noise and fewer texture details, which is not conducive to visual feature extraction and tracking. (2) How to solve initialization parameters by fusing event data and IMU measurement in the initial stage of the VIO system. Visual-inertial initialization is crucial for VIO, which can help the system obtain parameters required for state estimation, including scale, gravity direction, IMU bias, etc. However, for existing event-based VIO systems, there are no initialization methods that can effectively fuse event data and IMU measurement together to estimate initialization parameters.

To this end, firstly, for the challenge (1), we propose a novel direct event-based VIO method for better fusion of event data and IMU measurement. This VIO method can directly estimate the state of the system based on event inverse depth frames, Time-Surface (TS) images, and pre-integrated inertial measurement in a sliding window non-linear optimization framework without feature tracking. Secondly, for challenge (2), we propose a semi-joint event-inertial initialization method to improve the performance of event-based VIO state estimation. This initialization method can obtain an up-to-scale camera trajectory based on the event-only initialization (since the trajectory estimated by event-based stereo VO methods, such as ESVO [8], always has a scale error, so the trajectory here is deemed up-to-scale) and solve initialization parameters using the event-inertial initial optimization. Thirdly, referring to ESVO [8], we implement an event-based stereo VIO system based on the above two proposed methods, namely ESVIO. Finally, we evaluate ESVIO on the public datasets and the experimental results show that ESVIO achieves better performance in both accuracy and robustness in different scenarios. To conclude, our contributions can be summarized as below:We present a direct event-based stereo VIO method for the first time, which uses the sliding window tightly coupled optimization method to directly estimate the optimal state of the system based on the event and IMU data without feature tracking.We propose a semi-joint event-inertial initialization method to estimate initialization parameters, including scale, gravity direction, initial velocities, accelerometer, and gyroscope biases, in two steps (event-only initialization and event-inertial initial optimization).We implement the event-based stereo VIO system, ESVIO, in C++ and evaluate it on four public datasets [25,26,27,28]. The results demonstrate that our system achieves good accuracy and robust performance when compared with the state-of-the-art, and, at the same time, with no compromise to real-time performance.

The rest of the paper is organized as follows. Section 2 gives a brief review of related works. The system overview of ESVIO is presented in Section 3, followed by the preliminaries in Section 4. The proposed event-inertial initialization method is introduced in Section 5, and the detailed direct event-based VIO pipeline for state estimation is given in Section 6. Section 7 provides experimental evaluations. Finally, our conclusion is drawn in Section 8.

## 2. Related Work

### 2.1. Visual-Inertial Odometry Methods

Considering different sensor fusion methods, VIO methods can be divided into loosely- and tightly coupled methods [29]. For the loosely-coupled method, firstly, it estimates the state of the system based on visual information and IMU measurement separately in two independent estimators, and, then, it solves the system state based on these two state estimation results. Loosely-coupled methods are common in the early stages of VIO development, generally through an Extended Kalman Filter (EKF) to achieve sensor fusion and the final state estimation, such as [30,31]. For the tightly coupled method, it directly estimates the system state based on visual information and IMU measurement together in one estimator. Compared with the loosely coupled method, the tightly coupled method needs to consider the interaction between different sensor data, and the algorithm implementation is more complicated, but it will obtain more accurate state estimation results. Until now, tightly coupled methods mainly achieve sensor fusion and state estimation through filter-based estimation methods (e.g., EKF) [14,32,33] or optimization-based estimation methods (e.g., graph optimization) [13,15,17,34,35,36,37].

Considering visual measurement processing, VIO methods can also be divided into two different paradigms: feature-based VIO methods and direct VIO methods. For the feature-based method, firstly, it extracts and tracks specific visual features from consecutive camera frames through feature descriptors. Then, it will minimize feature reprojection errors and IMU propagation errors to estimate the system state. At present, most of the existing VIO methods are feature-based methods due to their robustness and maturity, such as [13,15,33,38,39,40]. Unlike the feature-based method, the direct VIO method directly uses the intensity value of pixels to construct the photometric error between two camera frames, and then estimates the system state by minimizing photometric errors and IMU propagation errors without the procedure of feature extraction and tracking. Compared with feature-based methods, direct VIO methods have faster processing speed and better performance in textureless scenarios [41], such as [16,18,42,43]. However, direct methods typically require a higher frame rate than feature-based methods to maintain good performance [44].

### 2.2. Event-Based Visual Odometry and SLAM

For event-based VO/SLAM methods, they can be divided into monocular ones and stereo ones. Mueggler et al. [9] presented an onboard event-based monocular system for 6-Degrees of Freedom (DoF) localization with high speed and fast rotation. Kim et al., [10] can perform real-time 3D reconstruction, 6-DoF localization, and scene intensity estimation by EKF based on a monocular event camera. Kueng et al. [45] implemented a monocular VO system by event-based feature detection. Rebecq et al. [11] proposed EVO that is implemented by an image alignment tracker via semi-dense mapping. Bryner et al. [12] introduced a maximum-likelihood framework to estimate the camera motion. Nguyen et al. [46] used the Stacked Spatial LSTM Network to estimate camera poses with event images as input. However, monocular methods always face the scale consistency problem, and stereo methods perform better in this aspect thanks to the known depth information.

For event-based stereo VO/SLAM methods, Zhou et al. [8] proposed a stereo event-based VO system called ESVO. The system achieved semi-dense 3D reconstruction [25] and 3D–2D registration tracking [47] via inverse depth map and TS. Jiao et al. [26] analyzed and compared the performance of ESVO under different event representation methods. Compared with monocular methods, stereo ones allow for more accurate and robust state estimation. However, due to the hardware limitation of the event camera, it is still difficult to obtain accurate state estimation results for VO/SLAM methods based on the event camera only in some scenes [8]. Therefore, fusing more sensor information, such as IMU measurement, is a solution to improve the performance of event-based VO/SLAM methods.

### 2.3. Event-Based Visual-Inertial Odometry

Until now, there are not many VIO methods based on the event camera. Zhu et al. [19] presented the event-based VIO system, namely EVIO, to provide accurate metric tracking of a camera’s full 6-DoF pose. EVIO uses the sensor state and event information to track visual features within the event frame over time and fuses visual feature tracks with the IMU measurement to update the system state employing an EKF with a structureless vision model. Rebecq et al. [20] also proposed a feature-based method, but fused event and inertial measurements using a keyframe-based non-linear optimization framework to estimate camera motion. Based on [20], Vidal et al. [21] implemented the first VIO state estimation method that fuses events, standard frames, and inertial measurements in a tightly coupled manner to achieve accurate and robust state estimation. Mueggler et al. [22] proposed a VIO method with an event camera under the continuous-time framework which allows direct integration of the asynchronous events with micro-second accuracy and inertial measurements at high frequency. Gentil et al. [23] presented an event-based VIO method which uses line features for the first time and constructs different line feature constraints to assist in state estimation. Chen et al. [24] proposed an event-based stereo VIO pipeline with sliding windows optimization, and further extended it to include standard frames, which tightly integrated stereo event streams, stereo image frames, and IMU together, to achieve accurate feature-based state estimation.

However, most of the above event-based VIO methods (except for [24]) are designed for monocular systems, and most of them (except for [22]) belong to the category of the feature-based VIO method. Compared with the standard camera frame, the event image frame constructed from raw event data has the disadvantages of low resolution, high noise, and fewer texture details due to the working principle and hardware limitations of event cameras, which makes the event-based feature extraction and tracking not as accurate and stable as needed [44]. Therefore, our proposed ESVIO can obtain events’ depth by stereo constraint and adopt direct VIO method to implement the fusion of event data (event depth and event image frame) and inertial measurement without feature tracking.

### 2.4. Visual-Inertial Initialization Methods

To start a VIO system, some parameters need to be estimated in an initialization process, including scale, gravity direction, initial velocity, accelerometer and gyroscope biases, etc. A fast and accurate initialization is critical for the subsequent visual-inertial state estimation. Based on the classification of initialization in [48], we divide visual-inertial initialization methods into three categories: joint, disjoint, and semi-joint initialization.

Joint initialization is similar to the tightly-couple method, which directly fuses raw visual and inertial measurements and estimates all initialization parameters in one step. The joint initialization method was first proposed by Martinelli [49], who presented a closed-form solution to jointly solve feature depth (scale), gravity, accelerometer bias, and initial velocity by linear least squares. Subsequent joint initialization methods are basically extended based on this method, including [50,51,52]. The advantage of joint initialization is that the interaction between the parameters is fully considered, and the disadvantage is that the real-time performance and the convergence of the solution process are not stable.

Disjoint initialization is similar to the loosely coupled method, which processes visual and inertial information separately, and solves initialization parameters step by step. Disjoint initialization is based on the assumption that the up-to-scale camera trajectory can be estimated very accurately from pure monocular vision, and then use this trajectory to estimate the initialization parameters [48]. Disjoint visual-inertial initialization was pioneered by Mur-Artal et al. in ORB-SLAM-VI [35] and later adopted by Qin et al. in VINS-Mono [15,53]. All subsequent disjoint initialization methods are improvements to these two methods, such as [54,55,56]. Compared to joint initialization, disjoint initialization has faster solution speed and better robustness. However, due to insufficient consideration of the interaction between initialization parameters and possible visual measurement noise, disjoint initialization does not perform well in estimation accuracy.

Semi-joint initialization can be seen as a compromise between joint initialization and non-joint initialization. It first solves partial parameters based on the visual measurement and then estimates all initialization parameters in one step. This idea was first implemented by Yang et al. [34], but matured by Campos et al. in ORB-SLAM-3 [39,48]. In ORB-SLAM-3, visual-inertial initialization was formulated as two maximum a posteriori (MAP) estimations: visual-only MAP estimation responsible for obtaining up-to-scale camera poses and inertial-only MAP estimation responsible for solving all initialization parameters. Semi-joint initialization performs well in terms of real-time performance and accuracy performance.

In this paper, we design a semi-joint event-inertial initialization method for event-based VIO to achieve better VIO state estimation results.

## 3. System Overview of ESVIO

The system overview of ESVIO is shown in Figure 1. The proposed ESVIO can be divided into four components: the measurements processing unit, the initialization unit, the depth estimation unit, and the VIO state estimation unit.

The measurements processing unit is responsible for processing sensor measurements, including event processing and IMU measurement processing. For event processing, ESVIO converts event streams to TS images for subsequent depth estimation and state estimation (Section 4.2). For IMU measurements processing, ESVIO adopts the IMU pre-integration algorithm to handle IMU measurements which can efficiently reduce the complexity and calculation time cost of the IMU propagation model (Section 4.3).

The initialization unit is responsible for solving initialization parameters to improve the performance of VIO state estimation, including event-only initialization and event-inertial initial optimization. The first part will estimate an up-to-scale camera trajectory based on event data (Section 5.1). Additionally, the second part will solve initialization parameters by fusing event and IMU data in a sliding window optimization framework (Section 5.2).

The depth estimation unit is responsible for inverse depth frame construction. This unit estimates the single event’s depth by stereo event matching and fuses these asynchronous depth points to construct an inverse depth frame (Section 4.4).

The VIO state estimation unit is responsible for camera state estimation by fusion of event data and inertial measurement. Firstly, it selects a depth points subset to reconstruct the inverse depth frame (Section 6.1). Secondly, it generates negative TS images for state estimation (Section 6.2). Thirdly, it estimates the state of the VIO system (including the camera’s 6-DoF pose, velocity, and IMU biases) using a novel tightly coupled direct VIO pipeline in a sliding window non-linear optimization framework based on inverse depth frames, negative TS images and pre-integrated inertial measurement (Section 6.3).

## 4. Preliminaries

### 4.1. Notations

We denote ${}^{w}\left(\cdot \right)$ as the world frame. ${}^{b}\left(\cdot \right)$ is the body (IMU) frame. ${}^{c}\left(\cdot \right)$ is the camera frame, which is equivalent to the inverse depth frame used in ESVIO. In particular, we denote ${}^{cp}\left(\cdot \right)$ and ${}^{cp(r)}\left(\cdot \right)$ as the left and right camera image plane of the camera frame ${}^{c}\left(\cdot \right)$. Note that the camera frames used in the paper are all left camera frames. We use both rotation matrix R and quaternion q to represent rotation and use translation vector p to represent translation. We consider $x_{n}^{m}$ or $X_{n}^{m}$ as the vector x or matrix X from frame *n* to frame *m*. $m_{t}$ is the frame *m* with timestamp *t*. We also use $m_{i}$ to represent the frame *m* while taking the *i*-th inverse depth frame.

### 4.2. Event Presentation Based on Time-Surface

The output of the event camera is a stream of asynchronous events. We use $e_{k}=\left(u_{k},v_{k},t_{k},p_{k}\right)$ to represent the *k*-th event in event stream, containing the pixel coordinate $x_{k}\left(u_{k},v_{k}\right)\in R^{2}$, timestamp $t_{k}$ and polarity $p_{k}$. Referring to ESVO [8], ESVIO adopts an alternative event representation method called Time-Surface (TS). TS is a two-dimensional (2D) camera size map where each pixel stores the time information. At time *t*, it can be defined by
(1)$$I\left(x,t\right)≐exp\left(-\frac{t-t_{last}\left({}^{cp_{t}}x\right)}{\varphi }\right),$$
where $t_{last}$ means the timestamp of the latest event at the pixel ${}^{cp_{t}}x\left({}^{cp_{t}}u,{}^{cp_{t}}v\right)\in R^{2}$, and φ is the constant decay rate parameter (e.g., 30 ms).

Based on Equation (Equation 1), we can convert a stream of events to a TS image whose pixel’s intensity is the value of the corresponding pixel in the TS map re-scaled from [0, 1] to the range [0, 255].

TS can be considered as a kind of *fixed time window* event processing method. The length of the time window is determined by the TS image generation frequency and the average brightness (intensity) of the TS image is determined by φ in Equation (Equation 1). However, we find that the setting of φ with a fixed value cannot handle different camera motions (different occurrence frequencies of events), especially at a low speed. When the camera moves slowly (low event occurrence frequency), there are few pixels in the TS image that can provide enough intensity information for subsequent depth estimation and state estimation. In order to tackle this problem, we have made improvements to the TS method used in ESVO to cope with slow camera motion. Based on Equation (Equation 1), we count the number of events across the image plane in the latest time window as *n*, and define the improved TS value:(2)$$I\left(x,t\right)≐\left\{\begin{array}{ccc} exp(-\frac{t-t_{last}\left({}^{cp_{t}}x\right)}{\varphi }), & \; & n\geq N_{\tau }\\ exp(-\frac{t-t_{last}\left({}^{cp_{t}}x\right)}{\varphi_{N_{\tau }}}), & \; & n<N_{\tau } \end{array}\right.$$
where $N_{\tau }$ depends on the number of events for depth estimation and we set $N_{\tau }$ as *N* in Section 4.4, $\varphi_{N_{\tau }}$ is defined by
(3)$$\varphi_{N_{\tau }}\;=\;\frac{t-t_{N_{\tau }}}{t_{w}}\cdot \varphi ,$$
where $t_{N_{\tau }}$ is the timestamp of the latest $N_{\tau }$-th events across the image plane, and $t_{w}$ is the length of the time window of TS.

The improved TS also needs to be rescaled to the range [0, 255] to create the corresponding TS image. From Equations (Equation 2) and (Equation 3), we can find that: when events occur at a normal or high frequency (normal or fast camera motion, $n\geq N_{\tau }$), the improved TS will work as the TS used in ESVO; when events occur at a low frequency (slow camera motion, $n<N_{\tau }$), there are at least $N_{\tau }$ events which can provide enough intensity information in the TS image (when $t_{last}\left({}^{cp_{t}}x\right)=t_{N_{\tau }}$, the intensity of x’s corresponding pixel is $255\times exp(-t_{w}/\varphi )$) for depth and state estimation. Figure 2 shows the processes of different TS methods under slow camera motion in an office scene.

### 4.3. IMU Pre-Integration

IMU pre-integration [57] is a computationally efficient alternative to the standard inertial measurement integration. It performs the discrete integration of the inertial measurement dynamics in a local frame of reference, thus preventing the need to reintegrate the state dynamics at each optimization step [41]. In our ESVIO, we use IMU pre-integration methods proposed in [15] to process IMU measurement.

A low-cost IMU, generally including a 3-axis accelerometer and a 3-axis gyroscope, can measure the acceleration a and the angular velocity ω of the IMU. The raw measurements from IMU, $\hat{a}$ and $\hat{\omega }$, can be represented by
(4)$$\begin{array}{cc} {}^{b}\;\hat{a} & {=}^{b}\;\;a+R_{w}^{b}\;{}^{w}\;g{+}^{b}\;b_{a}{+}^{b}\;n_{a}\\ {}^{b}\;\hat{\omega } & {=}^{b}\;\;\omega {+}^{b}\;b_{g}{+}^{b}\;n_{g}, \end{array}$$
where ${}^{b}b_{a}$, ${}^{b}b_{g}$ and ${}^{b}n_{a}$, ${}^{b}n_{g}$ are the biases and additive noises from the accelerometer and the gyroscope in the IMU frame, respectively, ${}^{w}g$ is the gravity vector in the world frame. Referring to [15], we assume that the additive noises ${}^{b}n_{a}$ and ${}^{b}n_{g}$ are Gaussian, ${}^{b}n_{a}∼N\left(0,{\sigma_{a}}^{2}\right)$, ${}^{b}n_{g}∼N\left(0,{\sigma_{g}}^{2}\right)$. Acceleration bias and gyroscope bias are modeled as random walk, whose derivatives are Gaussian, $\dot{{}^{b}b_{a}}={}^{b}n_{b_{a}}∼N\left(0,{\sigma_{b_{a}}}^{2}\right)$, $\dot{{}^{b}b_{g}}={}^{b}n_{b_{g}}∼N\left(0,{\sigma_{b_{g}}}^{2}\right)$.

Given two IMU frames $b_{i}$ and $b_{i+1}$ that correspond to two consecutive inverse depth frames, position, velocity, and rotation in the IMU frame $b_{i}$ can be computed by integrating the kinematics equation of IMU-driven algorithm within the duration $t\in [t_{i},t_{i+1}]$:(5)$$\begin{array}{cc} R_{w}^{b_{i}}p_{b_{i+1}}^{w} & =R_{w}^{b_{i}}\left(p_{b_{i}}^{w}+{}^{w}v_{i}\Delta t-\frac{1}{2}{}^{w}\;g\Delta t^{2}\right)+\alpha_{b_{i+1}}^{b_{i}}\\ R_{w}^{b_{i}}\;{}^{w}v_{i+1} & =R_{w}^{b_{i}}\left({}^{w}v_{i}-{}^{w}\;g\Delta t\right)+\beta_{b_{i+1}}^{b_{i}}\\ q_{w}^{b_{i}}⊗q_{b_{i+1}}^{w} & =\gamma_{b_{i+1}}^{b_{i}}, \end{array}$$
where
(6)$$\begin{array}{cc} \alpha_{b_{i+1}}^{b_{i}} & =\int \;\;\int_{t\in [t_{i},t_{i+1}]}\left(R_{b_{t}}^{b_{i}}\left({}^{b_{t}}\;\hat{a}-{}^{b_{t}}\;b_{a}-{}^{b_{t}}\;n_{a}\right)\right)dt^{2}\\ \beta_{b_{i+1}}^{b_{i}} & =\int_{t\in [t_{i},t_{i+1}]}\left(R_{b_{t}}^{b_{i}}\left({}^{b_{t}}\;\hat{a}-{}^{b_{t}}\;b_{a}-{}^{b_{t}}\;n_{a}\right)\right)dt\\ \gamma_{b_{i+1}}^{b_{i}} & =\int_{t\in [t_{i},t_{i+1}]}\frac{1}{2}\Omega \left({}^{b_{t}}\;\hat{\omega }-{}^{b_{t}}\;b_{g}-{}^{b_{t}}\;n_{g}\right)\gamma_{b_{t}}^{b_{i}}dt, \end{array}$$
where
(7)$$\Omega \left(\omega \right)=\left[\begin{array}{cc} -{\left\lfloor \omega \right\rfloor }_{\times } & \omega \\ -\omega^{⊤} & 0 \end{array}\right],$$Δt is the duration between time interval $t\in [t_{i},t_{i+1}]$, ⊗ is the quaternion multiplication, ${\left\lfloor \cdot \right\rfloor }_{\times }$ is the skew-symmetric matrix of a 3D vector.

$\alpha_{b_{i+1}}^{b_{i}}$, $\beta_{b_{i+1}}^{b_{i}}$ and $\gamma_{b_{i+1}}^{b_{i}}$ in Equation (Equation 6) are called pre-integration terms. Pre-integration terms can be understood as the relative motion from IMU frame $b_{i}$ to $b_{i+1}$ without the change of the state of IMU frame $b_{i}$.

Based on IMU pre-integration terms in Equation (Equation 6), we can establish the linear Gaussian error state recursion equation of IMU measurements, and derive the equation corresponding covariance matrix and Jacobian matrix in continuous-time and discrete-time for state estimation referring to [15]. Finally, we write down the IMU measurement model which will be used to construct the IMU residual in the initialization method (Section 5) and the VIO method (Section 6):(8)$$\left[\begin{array}{c} {\hat{\alpha }}_{b_{i+1}}^{b_{i}}\\ {\hat{\beta }}_{b_{i+1}}^{b_{i}}\\ {\hat{\gamma }}_{b_{i+1}}^{b_{i}}\\ 0\\ 0 \end{array}\right]=\left[\begin{array}{c} R_{w}^{b_{i}}\left(p_{b_{i+1}}^{w}-p_{b_{i}}^{w}+\frac{1}{2}{}^{w}\;g\Delta t^{2}-{}^{w}v_{i}\Delta t\right)\\ R_{w}^{b_{i}}\left({}^{w}v_{i+1}+{}^{w}\;g\Delta t-{}^{w}v_{i}\right)\\ {\left(q_{b_{i}}^{w}\right)}^{-1}⊗q_{b_{i+1}}^{w}\\ {}^{b_{i+1}}b_{a}-{}^{b_{i}}b_{a}\\ {}^{b_{i+1}}b_{g}-{}^{b_{i}}b_{g} \end{array}\right].$$

### 4.4. Depth Estimation Based on Time-Surface

When stereo TS images are fed into the depth estimation unit, ESVIO will follow the depth estimation method proposed by ESVO [8] to estimate events’ depth and fuse an inverse depth frame with the same timestamp as the corresponding TS image. At time *t*, the depth of event $e_{t-\varepsilon }=\left({}^{cp_{t-\varepsilon }}x,t-\varepsilon ,p\right)$ (with $\varepsilon \in \left[0,\delta t\right],{}^{cp_{t-\varepsilon }}x\in R^{2}$) on the left image plane can be estimated based on *the stereo temporal consistency criterion across space-time neighborhoods of the events* presented in [8]. The depth estimation optimization objective function of $e_{t-\varepsilon }$ is defined by
(9)$$\begin{array}{c} {{}^{c_{t-\varepsilon }}\rho }^{★}=\underset{c_{t-\varepsilon }\rho }{\arg\min}C_{depth}\left({}^{cp_{t-\varepsilon }}x,{}^{c_{t-\varepsilon }}\rho ,I_{left}\left({}^{cp_{t}}\left(\cdot \right),t\right),I_{right}\left({}^{cp{\left(r\right)}_{t}}\left(\cdot \right),t\right),T_{c_{t-\delta t}}^{c_{t}}\right),\\ C_{depth}≐\sum_{{}^{cp_{t}}x_{1,i}\in {}^{cp_{t}}W_{1},{}^{cp{\left(r\right)}_{t}}x_{2,i}\in {}^{cp{\left(r\right)}_{t}}W_{2}}{r_{i}}^{2}\left({}^{c_{t-\varepsilon }}\rho \right), \end{array}$$
where ${}^{c_{t-\varepsilon }}\rho$ means the inverse depth of the event in the camera frame $c_{t-\varepsilon }$ at time t−ε, $I_{left}\left({}^{cp_{t}}\left(\cdot \right),t\right)$ and $I_{right}\left({}^{cp{\left(r\right)}_{t}}\left(\cdot \right),t\right)$ are the left and the right TS with the timestamp *t*, respectively, $T_{c_{t-\delta t}}^{c_{t}}=\left(R_{c_{t-\delta t}}^{c_{t}}|p_{c_{t-\delta t}}^{c_{t}}\right)$ is camera transformation matrix between camera frame $c_{t-\delta t}$ and $c_{t}$ computed from the VIO state estimation results, and the depth residual $r_{i}\left({}^{c_{t-\varepsilon }}\rho \right)$ is defined as
(10)$$r_{i}\left({}^{c_{t-\varepsilon }}\rho \right)≐I_{left}\left({}^{cp_{t}}x_{1,i},t\right)-I_{right}\left({}^{cp{\left(r\right)}_{t}}x_{2,i},t\right),$$
where ${}^{cp_{t}}x_{1,i}\in R^{2}$ and ${}^{cp{\left(r\right)}_{t}}x_{2,i}\in R^{2}$ are pixel coordinates on the event ($e_{t-\varepsilon }$) corresponding patches ${}^{cp_{t}}W_{1}$ and ${}^{cp{\left(r\right)}_{t}}W_{2}$ of the left and the right TS at time *t*, respectively.

Based on Equations (Equation 9) and (Equation 10), we can estimate the depth of the latest *N* events which occur at or before time *t*. In ESVIO, we set *N* as 1500 (*MVSEC* dataset [27]) and 1000 (*RPG* datasets [25] and *ESIM* dataset [26]) based on the corresponding setting in [8] for different sensor resolutions to obtain a trade-off result between accuracy performance and time cost. Then we will fuse these asynchronous event depth points into an inverse depth frame with timestamp *t* based on the *probabilistic model of estimated inverse depth* and *inverse depth filters* proposed in [8]. In the real implementation of ESVIO, we fuse the inverse depth frame with timestamp *t* based on 3N∼5N event depth points, where *N* event depth points are associated with timestamp *t* and the rest are associated with several timestamps before *t*. In this way, we can construct inverse depth frames containing historical depth information for a short period of time.

## 5. Event-Inertial Initialization Method

Initialization is important for VIO to obtain a series of parameters, such as scale, gravity direction, and IMU bias, etc. These initialization parameters can help the system align with the world coordinate system and the scale of the real world, improving the convergence speed and estimation accuracy of state optimization in the VIO system. In the traditional VIO field, the initialization method, which combines visual information and IMU measurement, has been developed maturely, such as the disjoint initialization method used in VINS-Mono [15,53], the semi-joint method used in ORB-SLAM-3 [39,48], etc. However, for the event-based VIO field, there is no initialization method that fuses event data and IMU measurement together to estimate initialization parameters.

Therefore, we present an event-inertial initialization method for ESVIO, which can estimate scale, gravity direction, accelerometer, and gyroscope biases in the initial stage of ESVIO. This method is a kind of semi-joint initialization method and performs well in terms of real-time performance and accuracy performance. Our initialization method can be split into two steps: event-only initialization and event-inertial initial optimization.

### 5.1. Event-Only Initialization

The initialization procedure of ESVIO starts with the event-only initialization to obtain an up-to-scale camera trajectory. This trajectory can provide translational and rotational constraints for the event-inertial initial optimization to improve the accuracy and time performance of the optimization.

Firstly, ESVIO will apply a modified Semi-Global Matching (SGM) method [58] provided in ESVO [8] to generate an initial inverse depth frame based on the stereo TS images. Secondly, ESVIO will execute ESVO’s tracking and mapping procedure bootstrapped by the initial inverse depth frame. At this period, ESVIO will maintain a sliding window that contains a series of up-to-scale camera poses estimated by the event-based stereo VO process (ESVO). The construction method of the sliding window can refer to Section 6.3. Note that, we find that the camera motion trajectory estimated by ESVO always with a scale error as 10∼15%, so the camera poses and the trajectory obtained here are called up-to-scale ones, and the subsequent event-inertial initial optimization also takes the scale factor as one of the initialization parameters. The up-to-scale camera poses are defined as ${\bar{T}}_{c_{i}}^{c_{0}}=\left[R_{c_{i}}^{c_{0}}|{\bar{p}}_{c_{i}}^{c_{0}}\right],i\in \left[0,n\right]$, where we set the first camera frame ${}^{c_{0}}\left(\cdot \right)$ in the sliding window as the reference frame and *n* is the length of the sliding window. Suppose we have the extrinsic parameters $\left[R_{c}^{b}|p_{c}^{b}\right]$ between the event camera (left) and the IMU, we can translate poses from camera frame to IMU frame by
(11)$$\begin{array}{cc} R_{b_{i}}^{c_{0}} & =R_{c_{i}}^{c_{0}}\cdot {R_{c}^{b}}^{⊤}\\ s{\bar{p}}_{b_{i}}^{c_{0}} & =s{\bar{p}}_{c_{i}}^{c_{0}}-{R_{c_{0}}^{b_{i}}}^{⊤}\cdot p_{c}^{b}, \end{array}$$
where $s\in R^{+}$ is the initialization parameter *scale* which can align the up-to-scale trajectory to the metric-scale trajectory. We will solve this parameter along with other initialization parameters in the next step.

### 5.2. Event-Inertial Initial Optimization

This step aims to solve an optimal estimation of the initialization parameters for ESVIO. Due to the mutual coupling between the initialization parameters, it is difficult to decouple all parameters and solve them sequentially in multiple steps. Therefore, we solve all initialization parameters through one-step optimization to obtain more accurate initialization results. When the camera motion in the sliding window maintained by the previous step (Section 5.1) is sufficient, ESVIO will carry out the event-inertial initial optimization using the sliding window optimization method based on the up-to-scale camera trajectory in the sliding window. We follow the method proposed in [53] to judge whether the camera motion is sufficient by checking the variation of pre-integration items $\alpha_{b_{i+1}}^{b_{i}}$ and $\beta_{b_{i+1}}^{b_{i}}$ in the sliding window.

Before event-inertial initial optimization, ESVIO will make a rough estimation of the gravity direction based on velocity pre-integration items $\beta_{b_{i+1}}^{b_{i}}$ in the sliding window. Considering two consecutive IMU frames $b_{i}$ and $b_{i+1}$ in the sliding window, the velocity pre-integration item in Equation (Equation 5) can be converted to the reference frame ${}^{c_{0}}\left(\cdot \right)$ as:(12)$$\begin{array}{c} R_{b_{i}}^{c_{0}}\beta_{b_{i+1}}^{b_{i}}=R_{b_{i}}^{c_{0}}\left({}^{b_{i}}v_{i+1}-{}^{b_{i}}v_{i}+{}^{b_{i}}g\Delta t\right), \end{array}$$
where $R_{b_{i}}^{c_{0}}$ can be obtained by Equation (Equation 11).

Adding all velocity pre-integration items as shown in Equation (Equation 12) we can obtain
(13)$$\overset{n-1}{\underset{i=0}{\sum }}R_{b_{i}}^{c_{0}}\beta_{b_{i+1}}^{b_{i}}={}^{c_{0}}v_{n}-{}^{c_{0}}v_{0}+{}^{c_{0}}gt\approx {}^{c_{0}}gt,$$
where *t* is the total duration of the sliding window.

When we ignore the camera velocity changes during the sliding window, we can obtain a rough estimation of the gravity direction by normalizing the result in Equation (Equation 13).

Based on the up-to-scale camera trajectory and the rough gravity direction, ESVIO will start the event-inertial initial optimization by the sliding window optimization method. The optimal variables in the sliding window are defined by
(14)$$\begin{array}{cc} X & ={\left[R_{w}^{c_{0}},s,x_{0},x_{1},\cdots ,x_{n}\right]}^{⊤}\\ x_{i} & =\left[{}^{b_{i}}v_{i},{}^{b_{i}}b_{a},{}^{b_{i}}b_{g}\right],i\in \left[0,n\right], \end{array}$$
where $R_{w}^{c_{0}}$ is the gravity direction such that the gravity in the camera frame ${}^{c_{0}}\left(\cdot \right)$ is expressed as ${}^{c_{0}}g=R_{w}^{c_{0}}\;{}^{w}g$, with ${}^{w}g={\left(0,0,G\right)}^{⊤}$ being *G* the Gravitational Constant, $x_{i}$ is the camera initial velocity and the IMU biases in body frame at the time while taking the *i*-th data frame. *n* is the number of data frames in the sliding window.

Then, the optimization cost function $C_{init}$ for the event-inertial initial optimization can be designed by
(15)$$\begin{array}{c} C_{init}=\underset{X}{\arg\min}\left\{\sum_{i\in B}∥r_{B}\left({\hat{z}}_{b_{i+1}}^{b_{i}},X\right){∥}_{P_{b_{i+1}}^{b_{i}}}^{2}+∥r_{p}{∥}_{P_{p}}^{2}\right\}, \end{array}$$
where $r_{B}\left({\hat{z}}_{b_{i+1}}^{b_{i}},X\right)$ is the residual for IMU measurement between consecutive IMU frames $b_{i}$ and $b_{i+1}$, $r_{p}$ is the prior residual for the accelerometer bias as $r_{p}=\sum ||{{}^{b_{i}}b_{a}}^{2}\left|\right|$ referring to the initialization method [48]: if the motion does not contain enough information to estimate $b_{a}$, this prior will keep its estimation close to zero; if the motion makes $b_{a}$ observable, its estimation will converge towards its true value. $P_{b_{i+1}}^{b_{i}}$ is the covariance matrix for the IMU measurement residual and can be computed iteratively with IMU propagation as proposed in [15]. $P_{p}$ is the covariance matrix for the prior residual and can be defined as the diagonal matrix of noise ${\sigma_{b_{a}}}^{2}$. As to the gyroscope bias $b_{g}$, it is observable from estimated camera orientations and gyroscope readings, so we do not need to set a prior factor for it [48].

For the IMU measurement residual $r_{B}\left({\hat{z}}_{b_{i+1}}^{b_{i}},X\right)$, it can be defined according to the IMU measurement model defined in Equation (Equation 8) and the up-to-scale camera poses defined in Equation (Equation 11) and has the same form as Equation (Equation 25) with different position, velocity and rotation residuals:(16)$$\begin{array}{cc} \delta \alpha_{b_{i+1}}^{b_{i}} & =R_{c_{0}}^{b_{i}}\left(s\left({\bar{p}}_{b_{i+1}}^{c_{0}}-{\bar{p}}_{b_{i}}^{c_{0}}\right)+\frac{1}{2}R_{w}^{c_{0}}\;{}^{w}\;g\Delta t^{2}-R_{b_{i}}^{c_{0}}\;{}^{b_{i}}v_{i}\Delta t\right)-{\hat{\alpha }}_{b_{i+1}}^{b_{i}}\\ \delta \beta_{b_{i+1}}^{b_{i}} & =R_{c_{0}}^{b_{i}}\left(R_{b_{i+1}}^{c_{0}}{}^{b_{i+1}}v_{i+1}+R_{w}^{c_{0}}\;{}^{w}\;g\Delta t-R_{b_{i}}^{c_{0}}\;{}^{b_{i}}v_{i}\right)-{\hat{\beta }}_{b_{i+1}}^{b_{i}}\\ \delta \gamma_{b_{i+1}}^{b_{i}} & =2{\left[{q_{b_{i}}^{c_{0}}}^{-1}⊗q_{b_{i+1}}^{c_{0}}⊗{\left({\hat{\gamma }}_{b_{i+1}}^{b_{i}}\right)}^{-1}\right]}_{xyz} \end{array}$$
where $R_{c_{0}}^{b_{i}}$($q_{b_{i}}^{c_{0}}$), $q_{b_{i+1}}^{c_{0}}$, ${\bar{p}}_{b_{i}}^{c_{0}}$ and ${\bar{p}}_{b_{i+1}}^{c_{0}}$ can be obtained from the up-to-scale camera trajectory estimated in the event-only initialization (Section 5.1), ${\left[\cdot \right]}_{xyz}$ means the real part of a quaternion which is used to approximate the three-dimensional rotational error.

The solution to the non-linear optimization problem in Equation (Equation 15) is similar to the state estimation in Equation (Equation 19), and a detailed description can be found in Section 6.3. Additionally, the Jacobian matrices of the IMU measurement residual can be found in Section A.1.

## 6. Direct Event-Based VIO Method

In this section, we propose a novel direct event-based VIO method for ESVIO to directly estimate camera state based on event data and IMU measurement without feature tracking. Firstly, the proposed method reconstructs the inverse depth frame to simplify the redundant depth information. Then, it generates the negative TS image for state estimation. Thirdly, based on inverse depth frames, negative TS images, and pre-integrated IMU measurement, the method estimates the optimal state of the system in a sliding window non-linear optimization framework. In the following, we will describe in detail the inverse depth frame reconstruction, the negative TS image generation, and the direct VIO state estimation.

### 6.1. Inverse Depth Frame Reconstruction

For ESVIO, the inverse depth frame provided by the depth estimation unit sometimes contains too many depth points with redundant depth information, which will bring additional computation overhead. To tackle this problem, we propose an inverse depth frame reconstruction method to filter the depth point subset according to the distribution of depth points and reconstruct the inverse depth frame based on these depth points. As illustrated in Figure 3, the method can be achieved following the below four steps:
*Step 1*: Project all depth points of the inverse depth frame to its corresponding TS image (same timestamp) to enhance the intensity information in this TS image.*Step 2*: Use EDLines [59] to extract line segments in the enhanced TS image to obtain the edge map (taking less than 1.5 ms per TS image in datasets used in Section 7).*Step 3*: Extract connected regions in the edge map based on the contour retrieval algorithm [60]. According to the connected regions and projected coordinates in the TS image, cluster the depth points into each connected region.*Step 4*: According to the proportion of the number of depth points in each connected region to the total number of depth points, determine the number of points to select in each connected region, and then randomly select this number of depth points from each connected region to reconstruct the inverse depth frame.

### 6.2. Negative TS Image Generation

In the TS image, the pixel with a large intensity value represents the recently triggered event. Additionally, in a certain direction of this pixel, the pixel’s intensity value will decrease as the distance increases, representing the same event triggered at multiple previous moments. Due to this characteristic, the TS image can be interpreted as a kind of anisotropic distance field which is usually used in edge-based VO systems [8]. Referring to ESVO [8], we invert the TS image to the negative one and utilize it to construct the minimization optimization problem for the VIO state estimation. The negative TS image can be created from a TS map by
(17)$$\bar{I}\left(x,t\right)=1-I\left(x,t\right),$$
then re-scale the pixel’s value from [0, 1] to the range [0, 255].

In ESVIO, the negative TS image will be used to construct event measurement residual and participates in the state estimation.

### 6.3. Direct VIO State Estimation Based on Event Data and IMU Measurement

For the direct method in the traditional VIO field, it constructs the visual measurement residual by the photometric error between two camera frames, which will utilize the whole image information for state estimation. However, the event can only reflect the brightness change, not the brightness intensity, and event-based VIO can hardly construct visual residual by the photometric error. Therefore, inspired by ESVO [8], for the direct method in ESVIO, we construct the event data residual based on the projection error of the inverse depth frame on the negative TS image, which can utilize the information of event inverse depth frames and event image frames (negative TS images) more comprehensively than feature-based methods.

After achieving the inverse depth frame reconstruction (Section 6.1) and the negative TS image generation (Section 6.2), based on the reconstructed inverse depth frame, ESVIO will construct *data frame* which contains the corresponding negative TS image with the same timestamp and pre-integrated IMU measurement between two consecutive inverse depth frames. The sliding window will consist of a fixed number of *data frames* as shown in Figure 4. Note that, referring to [8], we only use the left negative TS for VIO because incorporating the right negative TS does not significantly increase accuracy while it doubles the computation cost.

The full state variables in the sliding window while taking the *i*-th data frame are defined by
(18)$$\begin{array}{cc} X & ={\left[x_{0},x_{1},\cdots ,x_{n}\right]}^{⊤}\\ x_{i} & =\left[p_{b_{i}}^{w},{}^{w}v_{i},q_{b_{i}}^{w},{}^{b_{i}}b_{a},{}^{b_{i}}b_{g}\right],i\in \left[0,n\right], \end{array}$$
where $x_{i}$ is described as the system position, velocity, and orientation in the world frame at the time while taking the *i*-th data frame and the IMU biases in the corresponding IMU frame. *n* is the number of data frames in the sliding window.

We achieve the optimization of the state estimation based on inverse depth frames, negative TS images, and pre-integrated IMU measurement. All the state variables are optimized in the sliding window by minimizing the sum of the cost terms from all the measurement residuals, including IMU measurement residuals and event data residuals. We design the optimization cost function $C_{state}$ for state by
(19)$$\begin{array}{c} C_{state}=\underset{X}{\arg\min}\left\{\sum_{i\in B}∥r_{B}\left({\hat{z}}_{b_{i+1}}^{b_{i}},X\right){∥}_{P_{b_{i+1}}^{b_{i}}}^{2}+\sum_{j\in D,l\in T}\rho (∥r_{D,T}\left({\hat{z}}_{j}^{c_{l}},X\right){∥}_{P_{j}^{c_{l}}}^{2})\right\}, \end{array}$$
where $r_{B}\left({\hat{z}}_{b_{i+1}}^{b_{i}},X\right)$ is the residual for IMU measurement and B is the set of pre-integrated IMU measurement in the sliding windows. $r_{D,T}\left({\hat{z}}_{j}^{c_{l}},X\right)$ is the residual for event data. D and T are the sets of inverse depth frames and corresponding negative TS images in the sliding window, respectively. $P_{b_{i+1}}^{b_{i}}$ and $P_{j}^{c_{l}}$ are covariance matrices for the IMU measurement residual and the event data residual. $P_{b_{i+1}}^{b_{i}}$ can be computed iteratively with IMU propagation as proposed in [15], and $P_{j}^{c_{l}}$ is set as the identity matrix during the implementation of ESVIO. ρ is the Huber loss function [61], which is used to down-weight the large event data residual to improve the efficiency and reduce the impacts of noise events or incorrect depth points, defined as
(20)$$\rho (r)=\left\{\begin{array}{cc} r & ,r\leq k^{2}\\ 2k\sqrt{r}-k & ,r>k^{2} \end{array}\right.$$
where *k* is the scale of the Huber loss and we set *k* as 20 in ESVIO by trial-and-error method.

ESVIO uses Levenberg–Marquard (LM) algorithm to solve the minimization non-linear optimization problem in Equation (Equation 19). The optimal state vector X will be found by iteratively minimizing the Mahalanobis distance of all the residuals. At each LM iteration step, the state increment δX can be solved by following equation based on Equation (Equation 19):(21)$$\left(H_{B}+H_{D,T}\right)\delta X=\left(b_{B}+b_{D,T}\right),\delta X={\left[\delta x_{0},\delta x_{1},\cdots ,\delta x_{n}\right]}^{⊤},$$
where $H_{\left(\cdot \right)}$ is the Hessian matrix of residuals vector $r_{\left(\cdot \right)}$, we have $H_{\left(\cdot \right)}=J_{\left(\cdot \right)}^{⊤}P_{\left(\cdot \right)}^{-1}J_{\left(\cdot \right)}$ and $b_{\left(\cdot \right)}=-J_{\left(\cdot \right)}^{⊤}P_{\left(\cdot \right)}^{-1}r_{\left(\cdot \right)}$, where $J_{\left(\cdot \right)}$ is the Jacobian matrix of residuals vector $r_{\left(\cdot \right)}$ with respect to X, and $P_{\left(\cdot \right)}$ is the covariance matrix of measurements.

For position, velocity, and bias items of state vector X, the update operator and increments at each LM iteration step can be defined as
(22)$$p^{\prime }=p+\delta p,v^{\prime }=v+\delta v,b^{\prime }=b+\delta b.$$

For the rotation item of state vector X, since the four-dimensional rotation quaternion q is over-parameterized, we use a pertubation $\delta \theta \in R^{3}$ as the rotation increment referring to [15]. Therefore, the update operator of q at each LM iteration step can be defined as
(23)$$q^{\prime }\approx q⊗\left[\begin{array}{c} 1\\ \frac{1}{2}\delta \theta \end{array}\right].$$

We can also write the Equation (Equation 23) as the rotation matrix form:(24)$$R^{\prime }\approx R\left(I+{\left\lfloor \delta \theta \right\rfloor }_{\times }\right).$$

Considering the real-time performance, we do not set a specific marginalization step to marginalize old states. Formulations of measurements’ residuals will be defined in the following, and the Jacobian matrices can be found in Section A.2.

For the IMU measurement residual between two consecutive frames $b_{i}$ and $b_{i+1}$ in the sliding window, according to the IMU measurement model defined in Equation (Equation 8), it can be defined by
(25)$$r_{B}\left({\hat{z}}_{b_{i+1}}^{b_{i}},X\right)=\left[\begin{array}{c} \delta \alpha_{b_{i+1}}^{b_{i}}\\ \delta \beta_{b_{i+1}}^{b_{i}}\\ \delta \gamma_{b_{i+1}}^{b_{i}}\\ \delta b_{a}\\ \delta b_{g} \end{array}\right]=\left[\begin{array}{c} R_{w}^{b_{i}}\left(p_{b_{i+1}}^{w}-p_{b_{i}}^{w}+\frac{1}{2}{}^{w}\;g\Delta t^{2}-{}^{w}v_{i}\Delta t\right)-{\hat{\alpha }}_{b_{i+1}}^{b_{i}}\\ R_{w}^{b_{i}}\left({}^{w}v_{i+1}+{}^{w}\;g\Delta t-{}^{w}v_{i}\right)-{\hat{\beta }}_{b_{i+1}}^{b_{i}}\\ 2{\left[{q_{b_{i}}^{w}}^{-1}⊗q_{b_{i+1}}^{w}⊗{\left({\hat{\gamma }}_{b_{i+1}}^{b_{i}}\right)}^{-1}\right]}_{xyz}\\ {}^{b_{i+1}}b_{a}-{}^{b_{i}}b_{a}\\ {}^{b_{i+1}}b_{g}-{}^{b_{i}}b_{g} \end{array}\right].$$

For the event data residual between two data frames, inspired by ESVO [8], it can be defined as the depth projection error of one’s inverse depth frame on another’s negative TS image. Considering the *j*-th inverse depth frame and the *l*-th negative TS image (j<l) in the sliding window, the event data residual can be defined by
(26)$$\begin{array}{cc} r_{D,T}\left({\hat{z}}_{j}^{c_{l}},X\right) & =\sum_{m\in D_{j}}{\bar{I}}_{left}\left(\left({}^{cp_{l}}u_{m},{}^{cp_{l}}v_{m}\right),t_{l}\right)=\sum_{m\in D_{j}}{\bar{I}}_{left}\left(\pi \left({}^{c_{l}}\;P_{m}\right),t_{l}\right),j<l\\ {}^{c_{l}}\;P_{m} & =\left[\begin{array}{c} {}^{c_{l}}\;x_{m}\\ {}^{c_{l}}\;y_{m}\\ {}^{c_{l}}\;z_{m} \end{array}\right]=R_{b}^{c}\left(R_{w}^{b_{l}}\left(R_{b_{j}}^{w}\left(R_{c}^{b}\left({}^{c_{j}}\;P_{m}\right)+p_{c}^{b}\right)+p_{b_{j}}^{w}-p_{w}^{b_{l}}\right)-p_{c}^{b}\right), \end{array}$$
where ${}^{c_{j}}\;P_{m}\in R^{3}$ is the *m*-th depth point in the *j*-th inverse depth frame, ${}^{c_{l}}\;P_{m}\in R^{3}$ is the same depth point but in the *l*-th camera frame coordinate system. Additionally, $\left({}^{cp_{l}}u_{m},{}^{cp_{l}}v_{m}\right)$ is the projected pixel location of ${}^{c_{l}}\;P_{m}$ in the image plane of the *l*-th negative TS image. π is the projection function that turns a 3D point to a 2D pixel location using event camera intrinsic parameters.

## 7. Experimental Evaluation

We implement the ESVIO system referring to the implementation of the ESVO system [8] based on Robot Operating System (ROS) [62] in C++. In the ESVIO system, there are three different independent threads, namely *time_surface* thread, *depth_estimation* thread, and *VIO* thread. They run concurrently to achieve the VIO state estimation. *time_surface* thread implements the conversion of the event stream into TS images, which is a part of the measurements processing unit (in Figure 1). *depth_estimation* thread implements the depth estimation unit (in Figure 1). *VIO* thread implements the pre-integration of IMU measurement (another part of the measurements processing unit shown in Figure 1), the initialization unit (in Figure 1), and the VIO state estimation unit (in Figure 1). Each thread has a different running rate accordingly to ensure reliable operation and the real-time performance of the whole system.

To evaluate the proposed ESVIO system, we perform corresponding quantitative and qualitative experimental evaluations on the public *RPG* dataset [25] and *MVSEC* dataset [27]. The data provided by the *RPG* dataset were collected with a handheld stereo event camera in an indoor environment. Additionally, the sequences we used for 6-DoF pose estimation from *MVSEC* dataset were collected using a stereo event camera mounted on a drone flying in a capacious indoor environment. We also evaluate the ESVIO on the *ESIM* dataset, which is generated by Jiao et al. [26] using the event camera simulator ESIM [63]. In addition, we try to evaluate ESVIO on the driving dataset *DSEC* [28] where the data were collected from a stereo event camera mounted on a driving car with a higher resolution. These four datasets all provide stereo event stream, IMU measurement, intrinsic parameters of event cameras, extrinsic parameters between event cameras and the IMU. The ground truth 6-DoF poses are also provided except *DSEC* dataset. However, none of these four datasets provide ground truth IMU bias, which can be used for the initialization evaluation. We list the parameters of sensors in each dataset in Table 1.

Based on the above datasets, firstly, we show the performance of the event-inertial initialization proposed in ESVIO. Secondly, we compare ESVIO with state-of-the-art methods in terms of accuracy performance. Thirdly, we show the improvement brought by the addition of IMU to ESVIO. Fourthly, we show the results of ESVIO on sequences from the driving dataset *DSEC*. Finally, we analyze the real-time performance of our system. Considering the randomness of the algorithm results [26], the results of quantitative evaluations (motion estimation, scale error, and time cost) are all average results of 10-trial tests. Additionally, considering the possible scale error in the adopted comparison methods, we align the estimated trajectory with ground truth using Sim(3) Umeyama alignment [64] in all evaluations. All experiments run on a computer equipped with Intel^®^ Core™ i9-10900K CPU @ 3.70 GHz and Ubuntu 20.04 LTS operation system.

### 7.1. Performance of the Event-Inertial Initialization

To prove the effectiveness of our proposed event-inertial initialization method for ESVIO, we compare ESVIO and its variant without the proposed event-inertial initialization (simply “ESVIO without initialization”) on the *RPG* and *MVSEC* datasets which are collected in the real world. The ESVIO without initialization is achieved by the direct VIO state estimation method provided in Section 6 with the initial guesses for gravity direction (solved by Equation (Equation 13)), accelerometer bias (zero), gyroscope bias (zero), velocities, and depth (recovered from the event-only initialization method in Section 5.1).

Because *RPG* and *MVSEC* datasets do not provide IMU calibration parameters and ground truth IMU bias, we choose the scale error and the mean absolute translational error (ATE) as the metric for evaluations. We compare the scale error at the beginning of the VIO system (5 s after the system starts running) and at the end of the system for ESVIO and ESVIO without initialization. The scale error at the beginning of the system can directly reflect the influence of initialization on scale recovery. The scale error at the end of the system and the mean ATE can represent the effect of initialization on the VIO state estimation. As shown in Table 2, ESVIO demonstrates lower scale errors and better accuracy performance than ESVIO without initialization in most sequences. The results in Table 2 show that the event-inertial initialization can effectively recover the scale and improve the accuracy performance for ESVIO state estimation. In addition, note that although the lack of initialization will make the VIO system have a scale error in the initial stage, due to the addition of IMU, the scale error will decrease after the system works for a while.

### 7.2. Evaluation of the VIO State Estimation Accuracy for ESVIO

To show the state estimation accuracy performance of ESVIO, we perform quantitative and qualitative evaluation experiments on the public *ESIM*, *RPG*, *MVSEC* datasets.

Firstly, the state estimation results of ESVIO evaluated on *ESIM*, *RPG*, and *MVSEC* datasets are shown in Figure 5. As shown in Figure 5, the trajectories (the fourth row) estimated by ESVIO have a good accuracy performance compared to the ground truth. Furthermore, the TS images (the second row) and inverse depth frames (the third row) from Figure 5 show the performance of ESVIO for event processing and depth estimation.

Secondly, we compare ESVIO with the state-of-the-art event-based stereo VO methods, including ESVO [8] and variants of ESVO with different event processing methods (refer to [26]), on *ESIM*, *RPG*, and *MVSEC* datasets. Table 3 shows the mean ATE under 12 sequences for ESVIO, ESVO, and variants of ESVO. The subscript of ESVO represents the different event-processing methods. TS represents TS (ESVO*_TS_* equals to the original ESVO). EM2000, EM3000, EM4000, EM5000 represent Event Map with 2000, 3000, 4000, 5000 events per map. EMTS represents a combination of TS and Event Map. We refer to [26] for the results of ESVO and variants of ESVO. As shown in Table 3, ESVIO demonstrates better accuracy performance in terms of motion estimation on all sequences compared with the state-of-the-art event-based stereo VO methods. In addition, we plot the absolute translational and rotational error statistics of ESVIO and ESVO in Figure 6. As shown in Figure 6, for absolute translational error, ESVIO performs better than ESVO on all sequences. As to the rotational error, ESVIO performs better than ESVO in both rotational error median and distribution on most sequences. From Figure 6, the experimental results demonstrate that the proposed ESVIO not only has the advantages of accuracy performance, but also has the advantages of robustness compared with the event-based stereo VO method.

Thirdly, we compare ESVIO with the state-of-the-art event-based VIO method. At present, event-based VIO methods [19,20,21,22,23] are all designed for monocular systems and evaluated on the public dataset [65] which only provides monocular event data. Among these methods, only Ultimate SLAM [21] is open source method which is a feature-based VIO method and provides different VIO modes based on different sensor combinations. Therefore, we compare ESVIO and Ultimate SLAM on the same sequences used in Table 3, and the results are shown in Table 4. When we run Ultimate SLAM, we use its all parameter settings except the sensor calibration parameters. The results of Ultimate SLAM are collected under “event (E) + IMU” and “event (E) + IMU + frame (Fr)” two modes. As shown in Table 4, the performance of Ultimate SLAM is not as good as ESVIO. In most sequences, Ultimate SLAM cannot correctly estimate the state due to the failure of feature tracking (features can be extracted, but cannot be tracked stably). In the remaining sequences, although state estimation can be performed, the accuracy performance of Ultimate SLAM is still not as good as ESVIO due to the large estimation error in the initial stage. From the results in Table 4, we can find that ESVIO has better accuracy and robustness performances compared with Ultimate SLAM.

### 7.3. Evaluation of the Impact of Adding IMU to ESVIO

To validate the improvement brought by the addition of IMU to ESVIO, we present the quantitative and qualitative experimental evaluations in this subsection. We evaluate ESVIO, ESVIO without IMU (state estimation implemented based on only event measurement residual), and ESVO on the public *ESIM*, *RPG*, and *MVSEC* datasets. The mean ATE and scale error are chosen as the evaluation metric. The evaluation results are shown in Table 5. The results show that ESVIO with IMU has more accurate state estimation accuracy and lower scale error, which demonstrates that the addition of IMU improves the state estimation accuracy performance for ESVIO. Additionally, we notice that ESVIO will not reduce to ESVO when no IMU measurement is used due to the different event processing methods (improved TS vs. TS) and different state estimation methods (sliding window optimization framework vs. 3D–2D registration between two consecutive frames). Note that, because [26] does not provide the scale error of ESVO, the results of ESVO in Table 5 are obtained by our own evaluation.

Then we choose two sequences *RPG_monitor_edited* and *MVSEC_indoor_flying1_edited* to show the details of translational and rotational errors of ESVIO and ESVIO without IMU measurement. The *RPG_monitor_edited* sequence contains more rotations, while the *MVSEC_indoor_flying1_edited* sequence is collected by a drone in a larger indoor space with a faster camera motion. As shown in Figure 7, the top subfigures are the estimated trajectories, the middle subfigures are the translational errors over time, and the bottom subfigures are the rotational errors over time. From the trajectories subfigures, we can discover that the trajectories estimated by ESVIO have a lower scale error, which are more consistent with the ground truth trajectories. From the translational error subfigures and the rotational error subfigures, we can intuitively find that ESVIO always has a lower value of the translational and rotational errors. Furthermore, from Figure 7, we can discover that the addition of IMU can significantly reduce rotation estimation error when the camera keeps rotating (e.g., 5–10 s in the *RPG_monitor_edited* sequence) or make a sharp turn (e.g., 3–7 s in the *MVSEC_indoor_flying1_edited* sequence).

### 7.4. Experiments on DSEC Dataset

To show the performance of ESVIO in large-scale scenes, we perform qualitative evaluation experiments on the public driving dataset *DSEC*. Driving scenarios are challenging for event-based sensors because forward motions typically produce considerably fewer events in the center of the image (where apparent motion is small) than in the periphery [66]. Additionally, the higher sensor resolution (640 × 480), the large-scale outdoor scenes and the dynamic objects (moving cars) are also challenging for ESVIO.

Since the *DSEC* dataset does not provide the ground truth 6-DoF poses, we only show the results of ESVIO for the *DSEC* dataset sequences *zurich_city_04* and *zurich_city_11_a* in Figure 8. As shown in Figure 8g–l, ESVIO can successfully achieve the camera pose estimation not only on the medium-length driving sequences ((g), (h), (j), (l)), but also on the long driving sequences of up to more than 300 m ((i), (k)). Furthermore, the reconstructed 3D maps and estimated trajectories from Figure 8m,n show that ESVIO can complete semi-dense depth estimation and state estimation in the case of sparse and dense events (events in *zurich_city_04_a* and *zurich_city_04_b* are denser than in *zurich_city_11_a*, due to the richer texture). However, because of the very heavy events load, ESVIO can not run in real-time on the *DSEC* dataset, so we slow down the playback of the *rosbag* when performing the corresponding experiments in this subsection.

### 7.5. Real-Time Performance Analysis

In this section, we analyze the real-time performance of our proposed ESVIO system.

Firstly, we present the time cost of the initialization of ESVIO. For the event-only initialization, ESVIO will take about 12 ms to execute the modified SGM method to generate an initial inverse depth frame. Then ESVIO will take about 10 ms and 46 ms to execute the tracking and mapping procedure of ESVO, respectively, for estimating up-to-scale camera poses. For event-inertial initialization, ESVIO will take less than 1 ms for the rough estimation of the gravity direction and about 95 ms for the event-inertial initialization optimization in the slide window optimization framework (including 10 data frames). Considering the visual observation conditions required for the modified SGM method and the collection and accumulation time of the data frames in the sliding window, the initialization will be completed within 2 s after the ESVIO starts.

Secondly, we show the real-time performance of ESVIO. Table 6 lists the detailed execution times of different threads of ESVIO under *MVSEC_ indoor_ flying1_ edited* sequence. As shown in Table 6, for ESVIO, the *time_surface* thread takes about 12 ms (∼ 83 Hz) to create the TS image, the *depth_estimation* thread takes about 46 ms (∼ 22 Hz) to estimate 1500 events’ depth and fuse 6000 depth points to the inverse depth frame, the *VIO* thread takes about 24 ms (∼ 42 Hz) to perform state estimation in the slide window optimization framework (including 6 data frames) based on 500 depth points per reconstructed inverse depth frame.

Regarding the choice for the size of the sliding window (6 as mentioned above), we justify it by showing its influence on the state estimation accuracy and computation cost. As shown in Table 7, we have investigated the effect of sliding window size (4, 6, 8, and 10 frames, respectively) in terms of state estimation accuracy and computation cost of the *VIO* thread under *MVSEC_ indoor_ flying1_ edited* sequence. The results show that the sliding window with 4 data frames presents the worst performance in accuracy, and the sliding window with 6/8/10 data frames demonstrates similar accuracy performance. However, the sliding window with 8/10 data frames has a larger computation cost. Considering the real-time performance, we set the sliding window size to 6 in the state estimation.

Regarding the choice for the number of depth points selected in the state estimation (500 as mentioned above), we implement the same experiments under *MVSEC_ indoor_ flyin- g1_ edited* sequence. Table 8 shows the state estimation accuracy and computation cost of the *VIO* thread with selecting 300, 500, 1000, and 1500 depth points per reconstructed inverse depth frame. As shown in Table 8, while increasing the number of depth points can effectively improve the state estimation accuracy, however, it also brings more computation cost. Considering the real-time performance, we choose to select 500 depth points per reconstructed inverse depth frame in the state estimation.

Regarding the number of events for depth estimation and the number of the depth points for depth fusion in the *depth_estimation* thread (1500 and 6000 as mentioned above), we refer to the corresponding settings in ESVO [8] and make some adjustments on this basis to obtain inverse depth frames with more depth information on different datasets.

## 8. Conclusions

This paper proposes a novel event-based stereo VIO system, namely ESVIO. Firstly, we present a direct event-based stereo VIO pipeline for the first time, which can directly estimate camera motion based on event data and IMU measurement in a tightly coupled sliding window non-linear optimization framework. Secondly, we design a semi-joint event-inertial initialization method that can solve initialization parameters in two steps at the initial stage of the VIO system. Based on the VIO and the initialization methods, we implement the ESVIO system. Corresponding experimental evaluations are conducted to prove the effectiveness of the proposed system, and the results show that ESVIO achieves good accuracy and robustness performance when compared with other event-based monocular VIO and stereo VO systems, and, at the same time, with no compromise to real-time performance.

However, ESVIO still has some limitations. Firstly, the performance of ESVIO is still limited by the hardware of event cameras. Secondly, ESVIO lacks the long-term data association mechanism, such as loop closure, which enables relocalization and drifts elimination.

In the future, we will introduce the standard camera frame to enhance the performance of ESVIO and implement loop closure for ESVIO. We will also implement a specific marginalization step for ESVIO. We would also like to record an event camera dataset that is more suitable for event-based VIO systems and event-inertial initialization methods to test and evaluate.

## Figures and Tables

**Figure 1 sensors-23-01998-f001:**
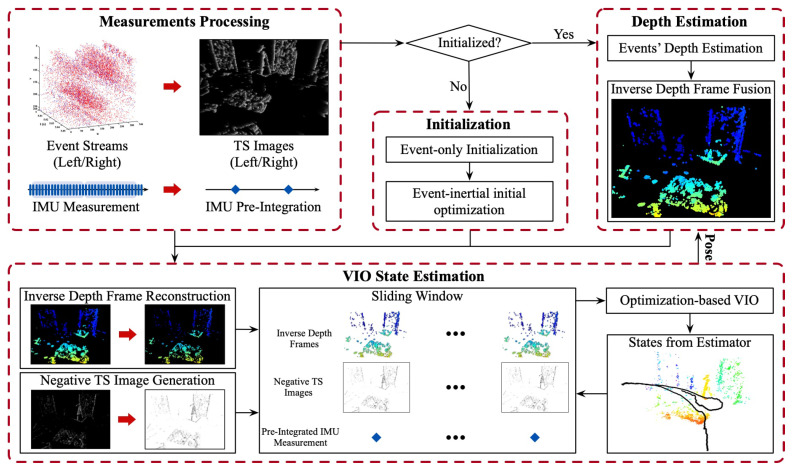
System overview of ESVIO. The measurements of stereo event cameras and IMU will be handled by the Measurements Processing Unit, and then the processed data will be sent to the Initialization Unit for event-only initialization and event-inertial initial optimization. After initialization, the Depth Estimation Unit will solve events’ depth and fuse the inverse depth frame. Finally, the VIO State Estimation Unit will estimate the camera’s 6-DoF pose, velocity, and IMU biases based on inverse depth frames, negative TS images, and pre-integrated IMU measurement.

**Figure 2 sensors-23-01998-f002:**
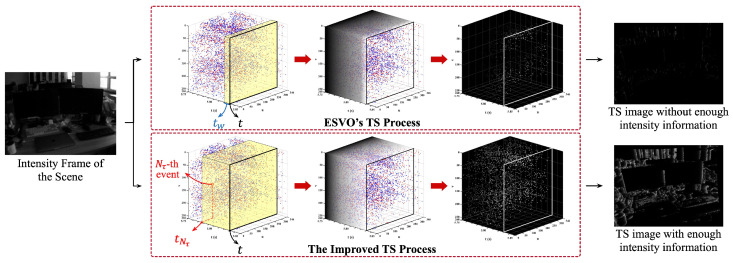
Different TS methods under slow camera motion in an office scene. When the camera is in slow motion, the improved TS will change the decay rate parameter based on $t_{N_{\tau }}$, while the ESVO’s will not. The left column is the intensity frames from the DAVIS346 event camera of the scene. The middle three columns are the TS processes: the first column shows the determination of $t_{N_{\tau }}$ and the events which can provide enough intensity information for TS images (in the yellow boxes); the second and third columns show how events are converted into pixels of different intensities under fixed φ (ESVO’s TS) and unfixed $\varphi_{N_{\tau }}$ (improved TS). The right column is the output TS images.

**Figure 3 sensors-23-01998-f003:**
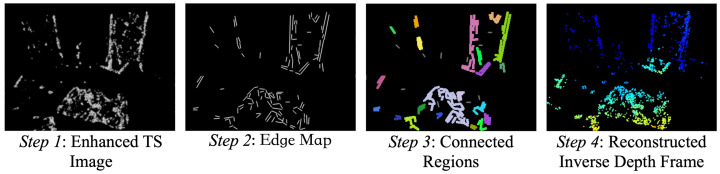
Four steps of the inverse depth frame reconstruction. Step 1: Enhance the TS image based on the inverse depth frame. Step 2: Extract the edge map from the enhanced TS image. Step 3: Extract connected regions in the edge map and cluster depth points. Step 4: Select depth points from each connected region according to the distribution of depth points to reconstruct the inverse depth frame.

**Figure 4 sensors-23-01998-f004:**
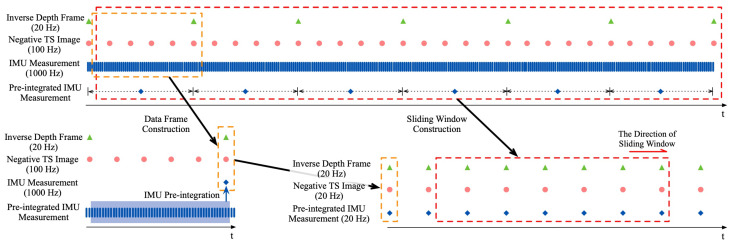
The sliding window construction of ESVIO. ESVIO construct *data frame* based on the frequency of inverse depth frames and construct the sliding window by a fixed number of *data frames*.

**Figure 5 sensors-23-01998-f005:**
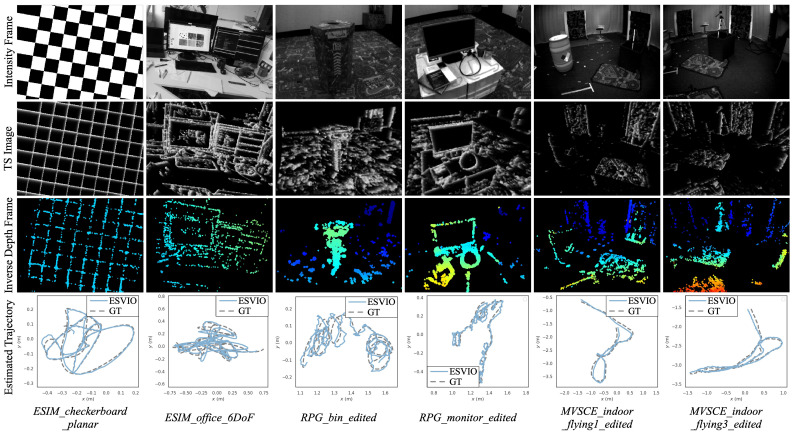
Results of ESVIO evaluated on *ESIM*, *RPG*, and *MVSEC* datasets. The first row shows intensity frames from the event camera, the second and the third rows show the TS images and the inverse depth frames generated by ESVIO, and the fourth row shows the trajectories estimated by ESVIO. Inverse depth frames are color-coded from red (close) to blue (far) over a black background.

**Figure 6 sensors-23-01998-f006:**
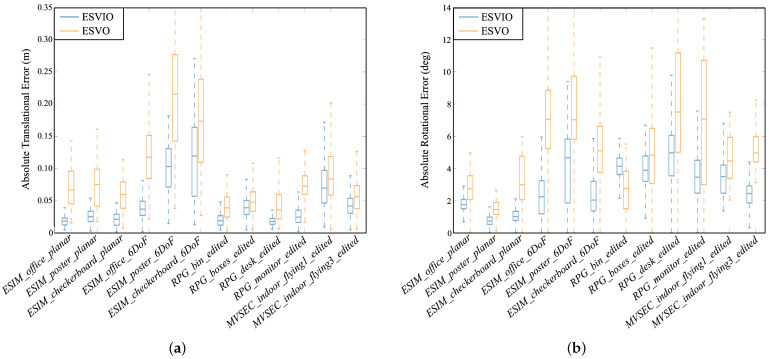
Boxplots of absolute translational (**a**) and rotational (**b**) error statistics for ESVIO and ESVO. The middle box spans the first and third quartiles, while the whiskers are the upper and lower limits. (**a**) Absolute Translational Error. (**b**) Absolute Rotational Error.

**Figure 7 sensors-23-01998-f007:**
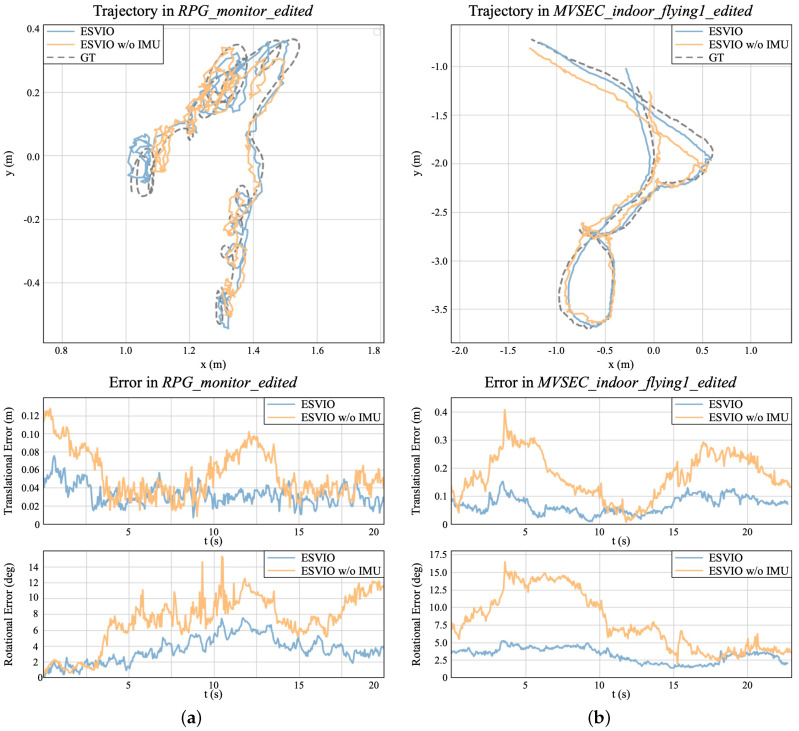
(**a**): Trajectories and the corresponding translational errors and rotational errors on the *RPG_monitor_edited* sequence. (**b**): Trajectory and the corresponding translational errors and rotational errors on the *MVSEC_indoor_flying1_edited* sequence. The blue line represents the results of the ESVIO, and the yellow line represents the results of ESVIO without IMU measurement. (**a**) *RPG_monitor_edited* sequence. (**b**) *MVSEC_indoor_flying1_edited* sequence.

**Figure 8 sensors-23-01998-f008:**
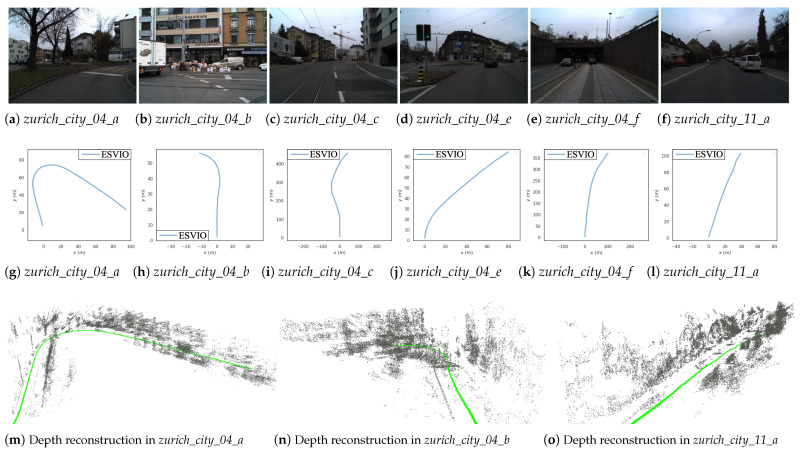
(**a**–**f**): Scene images of different sequences in *DSEC* dataset. (**g**–**l**): The trajectories produced by ESVIO with *DSEC* dataset sequences. (**m**–**o**): The reconstructed 3D maps and trajectories with *DSEC* dataset. The trajectories are in green and the depth points are in gray.

**Table 1 sensors-23-01998-t001:** Parameters of sensors in three datasets used in experimental evaluations. The IMU noise specs of MPU-6150 are from its product specification. The power spectral density is the square of noise amplitude density.

Dataset	Event Camera	Resolution (Pixel)	IMU	IMU Freq. (Hz)	IMU Noise Specs
*RPG* [25]	DAVIS240C	240 × 180	MPU-6150	1000	total RMS noise of gyr: 0.2${}^{\circ }$/s-rms power spectral density of acc: 400 μg/$\sqrt{Hz}$
*MVSEC* [27]	DAVIS346	346 × 260	MPU-6150	1000	
*ESIM* [26]	Simulator [63]	346 × 260	Simulator [63]	1000	gyr noise density: 1.86 × 10${}^{-4}$ rad/s$\sqrt{Hz}$ gyr random walk: 2.66 × 10${}^{-5}$ rad/s${}^{2}$$\sqrt{Hz}$ acc noise density: 1.86 × 10${}^{-3}$ m/s${}^{2}$$\sqrt{Hz}$ acc random walk: 4.33 × 10${}^{-4}$ m/s${}^{3}$$\sqrt{Hz}$
*DSEC* [28]	PPS3MVCD	640 × 480	MPU-9250	1000	gyr noise density: 5.07 × 10${}^{-3}$ rad/s$\sqrt{Hz}$ gyr random walk: 5.69 × 10${}^{-5}$ rad/s${}^{2}$$\sqrt{Hz}$ acc noise density: 7.81 × 10${}^{-2}$ m/s${}^{2}$$\sqrt{Hz}$ acc random walk: 1.33 × 10${}^{-3}$ m/s${}^{3}$$\sqrt{Hz}$

**Table 2 sensors-23-01998-t002:** Quantitative comparison results of ESVIO and ESVIO without initialization on the *RPG* and *MVSEC* datasets. The scale error and the mean absolute translational error (ATE) are used as the metric. The scale error includes the error at the beginning of the system (5s) and at the end of the system. The best result is highlighted in **bold**.

Sequence	Duration (s)	Method	Scale Error (%)		Mean ATE (cm)
			5s	End	
*RPG_bin_edited*	16.995	ESVIO	**1.8**	**0**	**2.0**
		ESVIO_w/o_initialization	8.2	0.1	2.2
*RPG_boxes_edited*	14.995	ESVIO	**8.7**	**2.2**	**4.0**
		ESVIO_w/o_initialization	15.8	4.0	4.7
*RPG_desk_edited*	12.995	ESVIO	**2.0**	**1.7**	**1.8**
		ESVIO_w/o_initialization	2.9	2.2	2.5
*RPG_monitor_edited*	22.985	ESVIO	**2.3**	**0.2**	**2.6**
		ESVIO_w/o_initialization	15.3	2.6	3.8
*MVSEC_indoor* *_flying1_edited*	25.976	ESVIO	**8.7**	**5.1**	**7.7**
		ESVIO_w/o_initialization	9.8	5.9	9.4
*MVSEC_indoor* *_flying3_edited*	24.984	ESVIO	**6.4**	3.8	**4.3**
		ESVIO_w/o_initialization	9.6	**3.6**	7.1
**Mean**	19.822	ESVIO	**5.0**	**2.2**	**3.7**
		ESVIO_w/o_initialization	10.3	3.1	5.0

**Table 3 sensors-23-01998-t003:** Accuracy performance with ESVIO and different event-based stereo VO methods. The public *RPG*, *MVSEC*, *ESIM* datasets are adopted for comparison. The mean ATE (cm) is used as the metric. The best result is highlighted in **bold** and the second best result is highlighted with underline.

Sequence	Duration (s)	Length (m)	ESVO${\;}_{{}_{T\;S}}$	ESVO${\;}_{{}_{E\;M2000}}$	ESVO${\;}_{{}_{E\;M3000}}$	ESVO${\;}_{{}_{E\;M4000}}$	ESVO${\;}_{{}_{E\;M5000}}$	ESVO${\;}_{{}_{E\;M\;T\;S}}$	ESVIO
*ESIM_ office_ planar*	15.999	5.014	4.7	4.0	3.9	3.7	4.1	4.9	**1.8**
*ESIM_ poster_ planar*	15.999	5.014	4.7	3.7	4.3	4.6	5.0	5.2	**2.5**
*ESIM_ checkerboard_ planar*	15.999	5.014	4.2	2.9	2.2	2.3	2.4	3.5	**2.0**
*ESIM_ office_ 6DoF*	29.999	15.003	9.1	25.3	21.0	16.6	15.8	9.4	**3.8**
*ESIM_ poster_ 6DoF*	29.999	15.003	18.2	15.4	16.3	16.8	17.4	17.8	**10.2**
*ESIM_ checkerboard_ 6DoF*	29.999	15.003	23.0	17.0	14.0	15.1	13.4	22.4	**11.5**
*RPG_ bin_ edited*	16.995	4.923	3.4	22.4	16.6	8.0	14.1	3.5	**2.0**
*RPG_ boxes_ edited*	14.995	6.686	6.5	5.3	17.1	13.7	9.8	22.1	**4.0**
*RPG_ desk_ edited*	12.995	3.926	3.4	2.9	3.3	3.2	2.9	3.6	**1.8**
*RPG_ monitor_ edited*	22.985	6.251	7.2	5.3	5.2	7.4	7.3	7.3	**2.6**
*MVSEC_ indoor_ flying1_ edited*	25.976	11.761	18.5	22.0	16.7	16.0	22.1	14.9	**7.7**
*MVSEC_ indoor_ flying3_ edited*	24.984	10.646	20.9	10.8	11.9	14.0	15.0	10.6	**4.3**
**Mean**	21.41	8.687	10.3	11.4	11.0	10.1	10.8	10.4	**4.5**

**Table 4 sensors-23-01998-t004:** Accuracy performance with ESVIO and Ultimate SLAM [21] on the *ESIM*, *RPG*, and *MVSEC* datasets. The mean ATE (cm) is used as the metric. The results of Ultimate SLAM are collected under “event (E) + IMU (I)” and “event (E) + IMU (I) + frame (Fr)” two modes. The symbol “-” means that the system can not correctly estimate states. The best result is highlighted in **bold**.

Sequence	ESVIO	Ultimate SLAM	Ultimate SLAM
		(E + I)	(E + I + Fr)
*ESIM_office_planar*	**1.8**	-	9.2
*ESIM_poster_planar*	**2.5**	5.8	7.3
*ESIM_checkerboard_planar*	**2.0**	-	7.0
*ESIM_office_6DoF*	**3.8**	-	-
*ESIM_poster_6DoF*	**10.2**	-	-
*ESIM_checkerboard_6DoF*	**11.5**	-	-
*RPG_bin_edited*	**2.0**	2.5	2.8
*RPG_boxes_edited*	**4.0**	-	-
*RPG_desk_edited*	**1.8**	-	-
*RPG_monitor_edited*	**2.6**	3.8	3.8
*MVSEC_indoor_flying1_edited*	**7.7**	-	-
*MVSEC_indoor_flying3_edited*	**4.3**	-	-

**Table 5 sensors-23-01998-t005:** Quantitative comparison results of ESVIO with and without IMU measurement on the *ESIM*, *RPG*, and *MVSEC* datasets. The scale error and the mean ATE are used as the metric. The best result is highlighted in **bold**.

Sequence	Scale Error (%)			Mean ATE (cm)		
	ESVIO	ESVIO_w/o_IMU	ESVO	ESVIO	ESVIO_w/o_IMU	ESVO
*ESIM_office_planar*	**2.4**	7.9	13.1	**1.8**	6.5	4.1
*ESIM_poster_planar*	**1.6**	12.4	11.0	**2.5**	6.7	4.7
*ESIM_checkerboard_planar*	**3.3**	10.7	13.0	**2.0**	5.3	4.9
*ESIM_office_6DoF*	**1.7**	11.6	24.3	**3.8**	9.1	11.5
*ESIM_poster_6DoF*	**2.0**	24.8	21.7	**10.2**	21.5	15.6
*ESIM_checkerboard_6DoF*	**10.7**	17.2	14.0	**11.5**	17.8	19.7
*RPG_bin_edited*	**0.0**	11.8	16.4	**2.0**	3.9	3.1
*RPG_boxes_edited*	**2.2**	11.1	24.7	**4.0**	5.0	8.8
*RPG_desk_edited*	**1.7**	6.7	7.7	**1.8**	3.2	3.6
*RPG_monitor_edited*	**0.2**	8.1	9.9	**2.6**	5.2	7.2
*MVSEC_indoor_flying1_edited*	**5.1**	9.5	12.7	**7.7**	13.1	14.1
*MVSEC_indoor_flying3_edited*	**3.8**	9.8	12.8	**4.3**	9.7	27.8
**Mean**	**2.9**	11.8	15.1	**4.5**	8.9	10.4

**Table 6 sensors-23-01998-t006:** Mean execution time of ESVIO’s different threads with *MVSEC_indoor_flying1* sequence of *MVSEC Dataset*.

Thread	Method	Times (ms)	Rate (Hz)
*time_surface*	Event processing	12.0	83.3
*depth_estimation*	Depth estimation	45.8	21.8
*VIO*	IMU pre-integration	0.4	41.5
	Data preprocessing	2.1	
	Nonlinear optimization	21.6	

**Table 7 sensors-23-01998-t007:** State estimation accuracy and computation cost of the *VIO* thread under different sizes of the sliding window optimization framework with *MVSEC_indoor_flying1* sequence. The mean ATE (cm) is used as the metric for accuracy comparison.

Size of Sliding Window	State Estimation Accuracy (cm)	Computation Cost of *VIO* Thread (ms)
4	10.2	16.9
6	7.7	24.1
8	7.5	36.5
10	8.3	44.2

**Table 8 sensors-23-01998-t008:** State estimation accuracy and computation cost of the *VIO* thread under different numbers of depth points per reconstructed inverse depth frame in the state estimation with *MVSEC_indoor_flying1* sequence. The mean ATE (cm) is used as the metric for accuracy comparison.

Number of Depth Points	State Estimation Accuracy (cm)	Computation Cost of *VIO* Thread (ms)
300	9.9	16.4
500	7.7	24.1
1000	7.8	43.8
1500	7.3	66.5

## Data Availability

Not applicable.

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
