# Peer review of "ESVIO: Event-Based Stereo Visual-Inertial Odometry"

_sensors, 2023, doi:10.3390/s23041998_

Round 1

Reviewer 1 Report

This paper presents a direct event camera-based stereo visual-inertial odometry method (ESVIO) built on top of an existing job, the event stereo visual odometry method (ESVO). The claimed novelty of the paper is the addition of an IMU to the event camera-based direct visual odometry system, whereas existing event camera-based visual-inertial odometry systems use a monocular camera or depend on feature extraction and tracking. In the proposed system, the novelty is epitomized by two improvements to ESVO: a joint optimization of state variables considering IMU factors in the initialization step, and a joint optimization of state variables considering IMU factors in the motion tracking step. So, to me, the novelty of the paper is limited.

Though the paper is easy to read, it is too verbose with lots of redundant descriptions (explained below), making a 26-page paper, and should be greatly condensed, maybe down to 20 pages.

In its present form, I would advise the authors to revise the paper thoroughly before resubmitting it for publication.

Major comments

1. Despite the limited novelty, the paper has contributions in terms of the addition of the IMU to a direct EVO system, and the numerous comparative experiments. But the experiments did not investigate the effect of sliding window size in terms of motion estimation accuracy and computation cost. I suggest adding such an experiment to strengthen the claimed contribution of the paper, the sliding window optimization-based motion tracking.

2. The paper is too wordy. For instance, please

a) shorten 4.1 notations, and 6.1 inverse depth frame reconstruction,

b) delete lines 296-302 because they are not used at all,

c) condense 6.1.2 negative TS image generation because this is proposed exactly in ESVO,

d) eq 17 and eq 23 are identical though the state variables are slightly different, please consider simplifying the equations.

e) condense 6.2.3 event data residual, since it is again adopted from ESVO.

f) Maybe delete line 393-394 and line 669-670?

3. Some important points of the methods are missing or unclear.

a) the description of the adaptive time window around eq 3 is confusing.

A1. “we count the number of events in the latest time window as n”: Does n only count the event in one pixel or across the image?

A2 “$t_N_{\tau}$ is the timestamp of the latest N_{\tau} events”: Does $t_N_{\tau}$ refer to the previous N_{tau}-th event at pixel x, or the previous N_{tau}-th event across the image?

A3 What is the physical meaning of t_w, “the length of the time window of TS”? How is it related to $\phi$ which, according to the text, is also the length of a time window?

b) line 320 “depth/inverse depth” should be “inverse depth”?

c) line 381, how to tell if the camera motion is sufficient?

d) in my opinion, in line 405, the prior factor on accelerometer biases is rather contrived. In my experience, for low-cost MEMS sensors like MPU6150, it is likely accel biases can reach 1 m/s^2, so this factor will definitely not “obtain a more accurate accel bias (line 408)”. So please be cautious in describing this factor. Also, why not use a prior factor on gyro biases?

e) In eq 18 and eq 24, the error state vector $\delta x_i$ is not defined anywhere in the text. Its explanation is necessary because the state variables involve SO3 elements.

f) line 459, “taking less than 1.5 ms” refers to one TS image?

g) In section 6.2.1, how does the optimization window slide? Is there a marginalization step as in VINS-Mono? My guess is that the optimization window does not involve any marginalization. The optimization sounds like the expanded motion tracking module of ESVO. Anyhow, please explain this point in section 6.2.1.

h) ESVO does not mention how to weight the event data residuals because it only uses the event data residuals and they can assume uniform weights. This situation is very different for the integration of event data residuals and IMU data residuals. Please explain how the event data residuals are weighted, i.e., how the covariance for the residuals is computed.

This is often crucial to successful nonlinear optimization.

i) Please clarify the up-to-scale trajectory in line 64 as has been done in lines 364-365. The seemingly contradictory up-to-scale notion and the stereo camera setup are baffling to a reader like me until lines 364-365 on page 10. So please explain it earlier.

Minor comments

1. Some unclear points in the experiments.

a) Please provide the IMU noise specs for MPU 6150 and the simulated data in ESIM, e.g., accel noise density, bias rank walk, etc.

b) line 590, please explain the ESVIO without initialization: This method has to be somehow initialized, right? how is this achieved?

c) line 640, “Ultimate SLAM cannot correctly estimate the state due to the failure of feature tracking.” This is difficult to understand. To me, Ultimate SLAM tracks features across both regular frames and event windowed frames. That is, in slow motion, tracking across regular frames should work fine. In fast motion, tracking across event frames should work. So, why did ultimate SLAM fail to track features in the tested sequences?

d) line 645, “the direct method is more suitable for event-based VIO”, this conclusion is a bit hasty. Why is not the performance advantage of the proposed method coming from stereo vision?

e) the experiments used ATE for evaluation, please explain what alignment is used in computing ATE. SE3 alignment or similarity alignment?

2. some grammatical issues

a) they all covert event stream --> they all convert …

b) IMU Bias, etc. --> IMU bias …

c) methof called Time-Surface --> method called …

d) line 399 is described the --> is the …

e) I suggest to use abbreviations or lowercase italics for TIME_SURFACE and DEPTH_ESTIMATION to save space.

f) line 590, “we compare the ESVIO with and without initialization”, please rephrase it, maybe “we compare the ESVIO and its variant without the proposed initialization method”.

g) line 628, robustness and stability --> robustness.

Reviewer 2 Report

The manuscript presents, to the best of my knowledge, the first direct event-based stereo visual-inertial odometry (VIO) method. The method builds on the direct method ESVO (TRO 2018), which tackles a similar problem but without fusing inertial measurements. The key contributions are the tightly-coupled fusion of stereo events and inertial measurements in an objective function for state estimation (NNLS problem in Eq (22)) and a novel initialization strategy to recover scale parameters, etc (NNLS problem in Eq (16)). I would also dare to say that the adaptive selection of the events in the time surface to deal with slow motions is novel (Eqs (2)--(3)). The method is tested on two standard, real-world datasets (RPG, MVSEC) and a simulated one (using the ESIM simulator). The method compares favorably against baseline methods, both stereo and monocular. The experiments also report the runtime capabilities of their implementation and the evaluation of the impact of adding IMU to the pipeline (ablation study).

Will the C++ code be released? I think I did not find a definite answer in the paper. It would be a valuable contribution.

Regarding the problem itself, there is concurrent work on feature-based event-based stereo VIO on arXiv, https://arxiv.org/abs/2212.13184 . Thus, the topic is timely. Several labs are making progress in this relevant research topic. While this reference is not cited and there is no need to cite it since arXiv is not peer-reviewed, I hope the authors consider it. I think a small textual comparison can only make the manuscript stronger. (Potentially revisit statements like L426).

The manuscript is well-written, organized and contains most of the relevant citations in the literature. It provides a good introduction to the topics, a proper categorization and a tutorial introduction to the techniques needed to tackle the problem.

In the comparison of feature-based vs direct methods (e.g. L115), it would be good if the discussion considered the frame rate, which is currently missing. Direct methods typically require higher rate than feature-based methods. That is, feature-based methods are more robust at lower frame-rates than direct methods; see, Fig 5 in Hidalgo-Carrió et al., Event-aided Direct Sparse Odometry, CVPR 2022.

Some comments by line number:

L1: standard cameras are not really treated in this manuscript. I do not think there is a need to mention them or the complementarity. I think it distracts the attention.

L33: what is this "hardware limitation"? (also in L135). Didn't event cameras offer advantages over standard cameras? I think this need to be better mentioned (could it be referring to what is written in L704? If so, move L704 earlier in the text, at the introduction, so that we know what it is referring to).

L44: all convert event streams to event image frames... -> I do not think this is true. For example, [22] does not convert events into images. Every event is treated in the objective function at its corresponding timestamp.

L71 (and L693): "We present a direct event-based VIO method for the first time" -> We present a direct event-based stereo VIO method for the first time. The method in [22] is monocular and also direct (no features are detected in the IJRR 2017 dataset). Consequently, the maps used are semi-dense, see Figs. 7 and 8 in [22].

L157: Related statement: "all of the above ... belong to the category of feature-based VIO" -> [22] is not feature-based. The experiments on IJRR 2017 show that it is direct, not extracting keypoints, resulting in semi-dense maps.

L159: On the disadvantages of high noise and less texture details to argue in favor of direct methods, similar arguments to justify the choice of direct methods are given in Hidalgo-Carrió et al., Event-aided Direct Sparse Odometry, CVPR 2022. See Section 2 "Why direct methods with events?" therein. It would be good to support the discussion with references.

L141: "[19] presented the first event-based VIO system". The "first" statements are always troublesome... [19] and [22] are concurrent "first" works. Just compare the arXiv time stamp of [22] (February 2017) with [19] (published in June 2017, CVPR - no arXiv version).

Section 6.1.2. When introducing the negative time surface (TS), I think this section should make reference to the contributions of ESVO, like Sec. 4.4 does. The negative TS was proposed in ESVO. The interpretation as an anisotropic distance field in L472, is also from the ESVO paper. Consequently, the design of the error in the introductory paragraph of Sec 6.2 should also refer to ESVO, since the negative TS was generated to define the camera tracking objective function, which is mimicked here in (26).

The statement in L503-505 about using only the event camera also appears in ESVO. Please cite or quote.

Section 7.3: I think it would make sense to write the ESVO numbers also in Table 5, and compare them. Even if ESVIO w/o IMU does not outperform ESVO, it's OK, because it is not its purpose to operate w/o IMU data. The important question for the reader is: does ESVIO reduce to ESVO when no IMU data is used?

Fig. 9: Please discuss how the alignment is performed. Typically, estimated and GT trajectories are aligned at t=0, and then they drift.

L686: How does runtime depend on these hyperparameters (6000, 500)? (Sensitivity analysis).

ESVO shows results on DSEC (YouTube video), albeit it is not real-time. Did the authors try to provide results on DSEC? It would make the paper stronger.

IMU data is provided, I believe.

L667: Please double check the motion in 5-10 s of monitor sequence. The small translation errors in Fig 9 do not mean that the camera "keeps rotating"... please revise taking a look at the actual motion in the sequences.

The derivatives / Jacobians: I think they could be moved to an appendix and/or better explain the notation. For example, the text says that they are derivatives with respect to xi (L417), but the formula writes with respect to delta-xi (Eq (18)). I think this is nowhere explained. Please also search for typos: Eq (18), I found a subscript (k+1) which should be (i+1).

Other:
L44: covert -> convert
L101: obtains -> obtain
L415: Equation (17) -> I think it meant to be (16).
L418: ...x_{i+1}, which are respectively...
L423: is respectively -> are respectively
L358: introduce the meaning of acronym SGM
L457: the manuscript emphasizes that it is direct, without feature extraction and (feature) tracking... but then L457 mentions there is some line feature extraction. Consider consistency of the whole text.
L617: origin ESVO -> original ESVO
L659: subfloiats -> subfigures
L669: additino -> addition

Round 2

Reviewer 1 Report

In my opinion, most of the reviewers’ questions have been addressed. The latest closely related publications have been cited. The quality of the paper has been greatly improved. However, I would recommend the authors address the below problems before publication.

1. Please explain the components of $\delta X$ in eq 21 or give some reference about how the error state is defined for Euclidean variables and SO3 variables like $R^w_{b_i}$.

2. Please explain the weight matrices for IMU factors and event camera residuals, and briefly describe how they are computed. E.g., $P_{b_{i+1}}^{b_i}$ and $P_p$ in eq 15, $P_j^{c_l}$ in eq 19.

3. line 152, why is the method in Chen et al [24] a ‘purely’ event-based stereo VIO pipeline as it also uses an IMU?

4. In section 4.1, maybe delete line 228;

Also, please rephrase line 236, “mt and mi are frame m with timestamp t and while taking the i-th inverse depth frame respectively.”, which is unclear.

5. In Table 1 of IMU noise specs, please correct the units for white noise density and random walk, referring to https://github.com/ethz-asl/kalibr/wiki/IMU-Noise-Model.

Also, please beautify the scientific notation, e.g., 1.86e-04 -> 1.86 $\times 10^{-4}$.

Plus, note that the power spectral density is the square of noise amplitude density (see https://en.wikipedia.org/wiki/Noise_spectral_density).

6. line 596, maybe delete “with Ultimate SLAM tracking method”

7. Grammatical problems

a. line 63, due to -> since

b. line 54, is called “up to scale” one -> is deemed up to scale

c. IMU datas -> IMU data, e.g., line 73, line 215

d. please rephrase line 153, “further extended it with image-aided, which tightly …” is confusing.

e. line 153, “sliding windows graph-based optimization” -> sliding window optimization?

f. line 622, discovery -> discover

g. line 634, 640 * 480 -> 640 $\times$ 480, also retouch the resolutions in Table 1.

h. Please proofread the paper thoroughly or use a tool like Grammarly to check the text. Many ‘a’ and ‘the’ are missing, and many plurals and possessives are improperly used.
